# Phytochemical Profiling and Bioactive Potential of Grape Seed Extract in Enhancing Salinity Tolerance of *Vicia faba*

**DOI:** 10.3390/plants13121596

**Published:** 2024-06-08

**Authors:** Doaa E. Elsherif, Fatmah A. Safhi, Prasanta K. Subudhi, Abdelghany S. Shaban, Mai A. El-Esawy, Asmaa M. Khalifa

**Affiliations:** 1Botany Department, Faculty of Science, Tanta University, Tanta 31527, Egypt; doaa.elsherif@science.tanta.edu.eg (D.E.E.); elesawy.mai@yahoo.com (M.A.E.-E.); 2Department of Biology, College of Science, Princess Nourah bint Abdulrahman University, Riyadh 11671, Saudi Arabia; faalsafhi@pnu.edu.sa; 3School of Plant, Environmental, and Soil Sciences, Louisiana State University Agricultural Center, Baton Rouge, LA 70803, USA; psubudhi@agcenter.lsu.edu; 4Botany and Microbiology Department, Faculty of Science (Boys), Al-Azhar University, Cairo 11884, Egypt; 5Botany and Microbiology Department, Faculty of Science, Al-Azhar University (Girls Branch), Cairo 11765, Egypt; asmaakhalifa2541.el@azhar.edu.eg

**Keywords:** salinity stress, alleviation of abiotic stress, natural product, antioxidant enzymes, photosynthetic activity, gene expression, grape seed

## Abstract

Salinity stress poses a significant threat to crop productivity worldwide, necessitating effective mitigation strategies. This study investigated the phytochemical composition and potential of grape seed extract (GSE) to mitigate salinity stress effects on faba bean plants. GC–MS analysis revealed several bioactive components in GSE, predominantly fatty acids. GSE was rich in essential nutrients and possessed a high antioxidant capacity. After 14 days of germination, GSE was applied as a foliar spray at different concentrations (0, 2, 4, 6, and 8 g/L) to mitigate the negative effects of salt stress (150 mM NaCl) on faba bean plants. Foliar application of 2–8 g/L GSE significantly enhanced growth parameters such as shoot length, root length, fresh weight, and dry weight of salt-stressed bean plants compared to the control. The Fv/Fm ratio, indicating photosynthetic activity, also improved with GSE treatment under salinity stress compared to the control. GSE effectively alleviated the oxidative stress induced by salinity, reducing malondialdehyde, hydrogen peroxide, praline, and glycine betaine levels. Total soluble proteins, amino acids, and sugars were enhanced in GSE-treated, salt-stressed plants. GSE treatment under salinity stress modulated the total antioxidant capacity, antioxidant responses, and enzyme activities such as peroxidase, ascorbate peroxidase, and polyphenol oxidase compared to salt-stressed plants. Gene expression analysis revealed GSE (6 g/L) upregulated photosynthesis (chlorophyll a/b-binding protein of LHCII type 1-like (*Lhcb1*) and ribulose bisphosphate carboxylase large chain-like (*RbcL*)) and carbohydrate metabolism (cell wall invertase I (*CWINV1*) genes) while downregulating stress response genes (ornithine aminotransferase (*OAT*) and ethylene-responsive transcription factor 1 (*ERF1*)) in salt-stressed bean plants. The study demonstrates GSE’s usefulness in mitigating salinity stress effects on bean plants by modulating growth, physiology, and gene expression patterns, highlighting its potential as a natural approach to enhance salt tolerance.

## 1. Introduction

Climate change presents a formidable challenge to global farming systems, and salinity stress is emerging as a major threat to crop productivity. Recent estimates by Kumar et al. [1] indicate that salinity stress affects approximately between one-quarter and one-third of the global land area. The Food and Agricultural Organization (FAO) further reports that over 424 million hectares of topsoil and 833 million hectares of subsoil, covering 85% of the global land area, are afflicted by salinity stress [2]. This phenomenon primarily results from irrigation practices using water with high salt content and the intrusion of saline water from the sea and rivers, particularly prevalent in arid and semiarid regions [3]. Factors including reduced rainfall and rising sea levels exacerbate soil salinization, compounding its detrimental effects on agricultural productivity [1].

Salinity stress adversely affects plants at multiple levels, encompassing morphological, physiological, biochemical, and molecular responses. High salinity levels disrupt photosynthetic activity, diminish shoot and root biomass, impair seed germination and seedling growth, and induce oxidative damage, electrolyte leakage, membrane disruption, stomatal closure, nutrient imbalance, decreased photosynthesis, and altered phytohormone production [4,5]. In response, plants must trigger various adaptive mechanisms, including enhancing phytohormone and osmoprotectant synthesis and upregulating antioxidant activities, to withstand salinity stress [6,7,8,9]. Faba bean is a legume considered moderately sensitive to salinity, exhibiting a reduction in plant growth of up to 50% under salinity stress [10]. One of the tolerance mechanisms underlying salt stress in faba bean is activated by antioxidant enzymes such as superoxide dismutase (SOD), peroxidase (POD), and polyphenol oxidase (PPO) activities, depending on genotype and salinity level [11].

Sustainable agricultural development seeks to minimize the reliance on inorganic fertilizers, conserve water resources, preserve biodiversity, and reduce waste production [12]. Exogenous ameliorants, particularly plant-based natural extracts, hold promise for enhancing plant resilience to environmental stresses, including salinity stress [13]. These extracts contain bioactive constituents such as amino acids, salicylic acid, and phenols, which bolster essential physiological processes, improve nutrient uptake efficiency, and ultimately enhance crop productivity independently of agrochemical inputs [14]. Notably, reports suggest that plants can bolster resilience to various abiotic stresses by harnessing organic extracts sourced from seaweed, maize grain, and licorice root [13,15]. In addition to grape seed extract, various other plant-derived extracts have been explored for their potential to mitigate salinity stress in crops. For instance, extracts from *Foeniculum vulgare* and *Ammi visnaga* seed improved the growth, yield, osmoprotectant content, antioxidant systems, water relations, photosynthetic efficiency, nutrient uptake, and K+/Na+ ratio of cowpea under salt stress [7]. Also, *Garcinia mangostana* pericarp extract increased plant height, leaf area, and yield components in mung bean under NaCl stress [16].

On the other hand, some studies used leaf extracts. *Moringa oleifera* and *Ocimum basilicum* leaf extracts mitigated oxidative stress and improved morphological and physiological parameters in common bean under 200 mM NaCl [17], as well as enhancing ion homeostasis, growth, photosynthetic pigments, organic solutes, phenols, and antioxidant enzymes in fenugreek under NaCl stress [18]. Also, *Moringa oleifera* leaf extract increased the cumulative yield and nutrient uptake in sorghum under different salinity levels [19].

Grape seed extract is a rich source of various compounds that possess potential health-promoting properties and diverse applications. Grape seeds are composed of significant amounts of protein, fiber, minerals, and water [20]. Additionally, they contain a considerable proportion of lipids, ranging from 7 to 20% [20]. Numerous studies have been conducted to investigate the bioactive constituents present in grape seeds [21,22]. Grape seed extracts are characterized by a heterogeneous mixture of monomeric, oligomeric, and polymeric forms of proanthocyanidins, which are the predominant compounds found in these extracts [23]. Proanthocyanidins constitute the major bioactive component in grape seed extracts [24]. These compounds are known for their antioxidant properties, which may contribute to alleviating stress, reducing inflammation, and promoting cardiovascular health [25]. Furthermore, grape seed extract has been suggested to support collagen synthesis and maintain elastin levels, thereby potentially benefiting skin health [26].

Faba bean (*Vicia faba* L.), a member of the Fabaceae family celebrated for its nitrogen-fixing capabilities and protein-rich seeds, faces formidable challenges from diverse abiotic stresses [27,28]. Particularly vulnerable to soil salinization, faba bean cultivation grapples with compromised growth and diminished yields. Despite escalating global demand, faba bean yields are dwindling due to environmental stressors, prominently soil salinity [18]. This study represents the first exploration into the application of grape seed extract (GSE) within agricultural practices. It aims to address the existing gap in our understanding of the physiological and genetic responses of *Vicia faba* L. (faba bean) plants to salinity stress. Furthermore, the efficacy of GSE as a potential ameliorative agent for mitigating salinity stress and enhancing crop resilience and productivity in *Vicia faba* L in saline conditions will be assessed.

## 2. Results

### 2.1. Identification of Phytochemical Components in Vitis vinifera Ethanolic Seed Extract

The phytochemical components present in GSE were meticulously identified through gas chromatography–mass spectrometry (GC–MS) analysis. The GC–MS spectral chromatogram depicting the analysis of grape ethanolic seed extract is represented in Figure 1, while a comprehensive breakdown of the bioactive components, including their retention time, molecular formula, and peak area percentage, is detailed in Table 1. The extensive data obtained from GC–MS spectroscopy unveiled several components, predominantly comprising fatty acids. Notably, the top ten major components discerned in the GSE are as follows: n-Hexadecanoic acid (26.4%), 9,12-Octadecadienoic acid (Z,Z)- (7.4%), Octadecanoic acid (6.6%), Hexadecenoic acid, Z-11- (3.8%), à-D-Glucopyranoside, O-à-D-glucopyranosyl-(1.fwdarw.3)-á-D-fructofuranosyl (3.1%), Linoleic acid ethyl ester (2.6%), Hexadecanoic acid, ethyl ester (2.3%), 2-Furancarboxaldehyde, 5-(hydroxymethyl)- (2.2%), Octanoic Acid (2.1%), and Hexanoic acid (1.7%).

### 2.2. Nutrient Content and Antioxidant Capacity of Grape Seed Extract

The analysis of critical nutrients supplemented with the applied grape seed extract revealed high levels of N, P, Ca, and Mg as common macronutrients and micronutrients, as shown in Table 2. In addition, the applied grape seed extract possessed high antioxidant capacity.

### 2.3. Effects of Grape Seed Extract on Growth Parameters

The treatment of faba bean seeds with 150 mM NaCl resulted in a significant reduction in shoot length, root length, fresh weight, and dry weight of 60-day-old bean plants by 30%, 52%, 25%, and 25%, respectively, compared to the control group (Figure 2 and Figure 3). Conversely, the foliar application of GSE notably enhanced all the measured growth parameters of salt-stressed bean plants. At GSE concentrations of 2, 4, 6, and 8 g/L, shoot length increased 86%, 97%, 111%, and 103%, respectively; root length increased 125%, 181%, 245%, and 191%, respectively; fresh weight increased 47%, 55%, 74%, and 39%, respectively; and dry weight increased 41%, 78%, 87%, and 70%, respectively, compared to salt-stressed plants. These findings highlight the potential of GSE to ameliorate the adverse effects of salinity stress on bean plant growth and development.

### 2.4. Effects of Grape Seed Extract on Photosynthetic Activity

The data presented in Figure 4 demonstrate that treating bean seeds with 150 mM NaCl resulted in a significant decrease in the Fv/Fm ratio of 60-day-old bean plants by 7% compared to the control group. In contrast, applying a foliar spray of grape seed extract at concentrations of 2, 4, 6, and 8 g/L enhanced the Fv/Fm ratio by 6%, 10%, 10%, and 7%, respectively, compared to plants under salinity conditions.

### 2.5. Effects of Grape Seed Extract on Oxidative Stress

Salinity stress notably increased the malondialdehyde (MDA) content of 60-day-old bean plants by 112% compared to non-salinized plants (Figure 5A). However, the foliar application of GSE at concentrations of 2, 4, 6, and 8 g/L resulted in reductions of the MDA content of salinized bean plants by 19%, 29%, 33%, and 4%, respectively. Similarly, salinity stress led to a significant increase in the hydrogen peroxide (H_2_O_2_) content by 47% in 60-day-old bean plants compared to non-salinized samples (Figure 5B). Conversely, foliar spray with grape seed extract at concentrations of 2, 4, 6, and 8 g/L decreased the H_2_O_2_ content of salinized bean plants by 12%, 22%, 39%, and 30%, respectively. Additionally, salinity stress induced a notable increase in the proline content by 48% compared to the control (Figure 5C). However, the foliar application of GSE at concentrations of 2, 4, 6, and 8 g/L decreased the proline content of salinized bean plants by 49%, 46%, 57%, and 43%, respectively. Moreover, salinity stress induced a notable increase in the glycine betaine content by approximately 30% compared with control samples (Figure 5D). Conversely, the foliar application of grape seed extract at all four concentrations decreased the glycine betaine content of salinized bean plants by approximately 10%, 40%, 70%, and 30%, respectively.

### 2.6. Effects of Grape Seed Extract on Osmolytes

Salinity stress resulted in a decrease in the content of total soluble proteins (TSPs) and total amino acids (TAAs) in 60-day-old bean plants by 36% and 17%, respectively, compared with the control group (Figure 6). Conversely, salinity stress led to a notable increase in the content of total soluble sugars (TSS) by 64% in the same plant age group compared with control samples. The application of grape seed extract via foliar spray at concentrations of 2, 4, 6, and 8 g/L increased the content of TSP, TAA, and TSS in salinized 60-day-old bean plants. The enhancements were observed as follows: 51%, 76%, 163%, and 76%, respectively, for TSP; 3%, 18%, 37%, and 8%, respectively, for TAA; and 4%, 76%, 88%, and 42%, respectively, for TSS. These findings suggest that grape seed extract may play a role in modulating osmolyte levels in bean plants under salinity stress conditions.

### 2.7. Effects of Grape Seed Extract on Antioxidants

Salinity stress significantly increased the total antioxidant capacity (TAC) of 60-day-old bean plants by 41% compared with control samples (Figure 7). Furthermore, the foliar application of 6 and 8 g/L GSE enhanced the TAC content of salinized 60-day-old bean plants by 38% and 29%, respectively. In terms of enzyme activities, treating bean seeds with 150 mM NaCl led to a notable increase in peroxidase (POD) and polyphenol oxidase (PPO) activities, while decreasing the activity of ascorbate peroxidase (APX), compared to the control group. Subsequently, foliar spray with grape seed extract at concentrations of 2, 4, 6, and 8 g/L improved the activities of POD and APX in salinized 60-day-old bean plants. Conversely, the activity of PPO in salinized bean plants was reduced by using foliar spray of GSE at concentrations of 2, 4, 6, and 8 g/L. This comprehensive analysis suggests that grape seed extract may play a significant role in modulating antioxidant responses and enzyme activities in bean plants under salinity stress conditions.

### 2.8. Effects of Grape Seed Extract on Relative Gene Expression

The impact of grape seed extract on the relative gene expressions of RbcL, Lhcb1, OAT, CWINV1, and ERF1 in 60-day-old bean plants was investigated (Figure 8). Salinity stress resulted in significant downregulations of RbcL, Lhcb1, and CWINV1, while concurrently inducing the expression levels of OAT and ERF1 in salinized 60-day-old bean plants compared with the control group. Conversely, the foliar application of GSE at various concentrations, particularly at 6 g/L, significantly upregulated the expression levels of RbcL, Lhcb1, and CWINV1 (Figure 8A,B,D). Notably, the same concentrations of GSE significantly downregulated the expression levels of OAT and ERF1 in salinized 60-day-old bean plants compared with their unstressed counterparts (Figure 8C,E). These results suggest that grape seed extract may play a role in modulating gene expression patterns associated with key physiological processes in bean plants under salt stress conditions.

### 2.9. Correlation Analysis of Physiological and Biochemical Traits

The correlation matrix analysis conducted under saline stress revealed significant relationships among various physiological and biochemical attributes in the bean plants (Figure 9). Shoot length exhibited strong positive correlations with root length (0.954), fresh weight (0.908), and dry weight (0.948). Similarly, root length showed positive associations with fresh weight (0.940) and dry weight (0.981). At the molecular level, a notable positive correlation was found between the expression of *RbcL* and *CWINV1* (0.995), indicating a potential link between photosynthesis and carbohydrate metabolism. Additionally, total soluble protein content positively correlated with root length (0.921), fresh weight (0.915), and dry weight (0.871). In contrast, negative correlations were observed between certain stress-related parameters and growth traits. Proline content negatively correlated with shoot length (−0.955), root length (−0.934), and fresh weight (−0.955). Polyphenol oxidase (PPO) activity exhibited negative correlations with glycine betaine (−0.922). Glycine betaine also showed negative correlations with shoot length (−0.690) and root length (−0.871). Hydrogen peroxide (H_2_O_2_) content demonstrated negative correlations with shoot length (−0.674) and root length (−0.841).

## 3. Discussion

Soil salinization significantly impacts global agriculture by adversely affecting plant morphology and physiology. Eco-friendly approaches, such as using natural plant extracts, have recently gained attention as alternatives for mitigating salinity stress. These extracts, known as plant bio-stimulants, contain essential phytochemicals, such vitamins, amino acids, carotenoids, minerals, phenolics, phytohormones, and antioxidants, which help alleviate salinity stress [29]. The use of grape seed extract as a natural approach to enhancing crop productivity in agricultural settings offers significant potential benefits, including cost-effectiveness, availability, and scalability. The practical applications of GSE are diverse and can contribute to improving the nutritional value and shelf life of agricultural products, while reducing waste and environmental issues associated with grape processing. The methods for administering these extracts include liquid foliar spray, root treatment, and soil incorporation [14]. Our study found a notable reduction in growth parameters, such as shoot length, root length, fresh weight, and dry weight, in 60-day-old bean plants under salinity stress (Figure 2). This reduction can be attributed to disrupted metabolic activities resulting from decreased water and mineral uptake [4,5]. The inhibition of cell elongation and division and impairment in various physiological and biochemical processes likely contribute to the observed reduction in plant biomass [30].

The GC–MS analysis of grape ethanolic seed extract revealed that the main component is n-Hexadecanoic acid (26.4%), a saturated fatty acid that inhibits the growth of soil-borne pathogens and enhances seedling growth [31]. It also plays a crucial role in the oxidative stability of oils [32]. 9,12-Octadecadienoic acid (Z, Z)-(7.442%) is an unsaturated fatty acid that acts as a strong antioxidant and a precursor to jasmonic acid, stimulating anti-stress responses [33]. Octadecanoic acid (6.6%) enhances the repair of Photosystem II by accelerating the de novo synthesis of D1 protein and mitigating the photo-inhibition of PSII [34]. α-D-Glucopyranoside (3.1%) acts as a signaling molecule regulating plant growth and antioxidant activities [35]. Linoleic acid ethyl ester (2.6%) functions as a plant growth regulator similar to gibberellic acid [36]. Octanoic Acid (2.1%) and Hexanoic acid (1.7%) possess antifungal properties, enhancing plant growth [37,38]. Analysis revealed that grape seed extract (GSE) contains significant levels of key macronutrients and micronutrients, contributing to plant health and resilience against stresses [39].

Our findings indicate that GSE application enhances all measured growth parameters of faba bean plants and alleviates the detrimental effects of salinity stress. This observation aligns with previous research reporting the stimulatory effects of various plant extracts [40]. Tomar et al. [41] found that plant extracts stimulate growth by upregulating enzyme activities in important metabolic pathways. The Fv/Fm ratio, an indicator of PSII efficiency, decreased under salt stress, indicating photosynthetic impairment [42]. However, GSE improved photosynthetic activity in salinized plants, as evidenced by an increase in the Fv/Fm ratio. This improvement is likely due to the enrichment of grape seed extract with nutrients that support the photosynthetic machinery and antioxidative defense system [43,44].

Salinity stress triggers the activity of reactive oxygen species (ROS), leading to cellular membrane disruption and oxidative damage [44,45]. However, GSE application reduced oxidative stress damage by scavenging H_2_O_2_, enhancing the activity of antioxidant enzymes like catalase (CAT), superoxide dismutase (SOD), and glutathione peroxidase (GPx) [46,47,48]. Proline and glycine betaine are crucial osmolytes that help plants cope with salinity stress by regulating osmotic balance and protecting cellular structures [49,50,51,52]. Our study showed an increase in proline and glycine betaine in response to salinity stress, but their levels significantly decreased after GSE application. This suggests that GSE, known for its antioxidant properties, may influence the levels of these osmolytes, potentially contributing to improved plant resilience [47].

Salinity stress is known to reduce the total soluble proteins (TSPs) and total amino acids (TAAs) of plants [53,54]. Our study demonstrated that GSE application significantly increased TSP and TAA levels in salinized bean plants, likely due to enhanced nutrient uptake and nitrogen assimilation [55,56]. Additionally, the increase in total soluble sugars (TSS) observed in salinized bean plants treated with GSE indicates improved osmo-protectant levels, which stabilize membrane integrity and mitigate stress effects [7,57,58].

The study also revealed that salinity stress increases the total antioxidant capacity (TAC) and peroxidase enzyme (POD) activity while decreasing the ascorbate peroxidase (APX) activity in bean plants, reflecting the plant’s defense mechanism against ROS accumulation [59,60,61]. GSE application stimulated antioxidant defenses, mitigating oxidative stress-induced damage [62]. Additionally, GSE reduced the polyphenol oxidase (PPO) activity, aligning with previous studies [63,64,65]. Their mechanism of action might involve phosphorus (P) release, the activation of nitrogen (N) metabolism, the stimulation of root growth, nutritional and hormonal regulation, or the generic stimulation of soil microbial activity. Previous reports documented that the application of biostimulants enhanced various physiological processes including plant nutrient uptake and utilization, photosynthesis, water use efficiency, the synthesis and concentration of growth hormones, germination, and senescence reduction [29,66], which in return increase plant production, yield, post-harvest quality, and the shelf life of agricultural products.

Salinity stress causes degradation of the light-harvesting complexes, impairs their function, and reduces electron transport, Rubisco activity, and chlorophyll absorption, thereby hindering carbon assimilation, as noted by [67]. Light energy is absorbed by photosynthetic antenna proteins. These proteins are further categorized into LHCA and LHCB, which assist PSI and PSII, respectively. PSII is a critical component involved in the photoreaction of photosynthesis, and Lhcb proteins also play a role in modulating plant responses to abiotic stress conditions [68,69]. LHCII trimers comprise proteins encoded by *Lhcb1*, *Lhcb2*, and *Lhcb3* genes [70]. The present study reveals that salinity stress suppresses *Lhcb1* expression. This is consistent with the findings of Xu et al. [71], who demonstrated that down-regulation or disruption of the *Lhcb1* gene reduces stomatal responsiveness to ABA, resulting in decreased tolerance to various stressors. However, the application of GSE, particularly at a concentration of 6 g/L, enhances *Lhcb1* expression, potentially leading to improved electron transport, Rubisco activity, chlorophyll absorption, and carbon fixation, thereby correlating with enhanced plant growth [67].

Ribulose-1,5-bisphosphate carboxylase/oxygenase (Rubisco) is a crucial enzyme in photosynthetic carbon assimilation, with Form I Rubisco in vascular plants consisting of a hexadecamer composed of eight large subunits (*RbcL*) encoded by the chloroplast genome and eight small, nuclear-encoded subunits (*RbcS*), as reported by [72]. It is believed that the expression level of the *RbcL* gene is modulated in response to various environmental conditions [73,74].

In this study, there was significant reduction in the expression level of the *RbcL* gene in 60-day-old salinized bean plants compared to control plants. The downregulation of the *RbcL* gene may hinder photosynthetic capacity by inhibiting carbon fixation [75]. However, the foliar application of different concentrations of GSE, especially 6 g/L, significantly induced the expression level of the *RbcL* gene, thereby enhancing photosynthetic activity [76]. This improvement in photosynthetic activity correlated directly with the increase in fresh weight of 60-day-old salinized bean plants, as evidenced by the Pearson correlation coefficient.

The *CWINV1* gene, responsible for encoding cell wall invertase, plays a crucial role in plants by regulating sugar metabolism and allocation. Cell wall invertases are enzymes essential for breaking down sucrose into glucose and fructose, thus influencing sugar balance and carbon distribution within the plant [77]. Extensive research has highlighted the significance of *CWINV1* in sugar signaling, growth, and developmental processes in plants [78,79]. Studies have indicated that *CWINV1* is not only pivotal in vascular anatomy and sugar accumulation but also contributes to plant stress adaptation mechanisms, particularly in response to environmental stresses such as salt stress [80]. Salt stress alters the expression levels of the *CWINV1* gene in plants. Under salt stress conditions, plants undergo changes in gene expression patterns to mitigate stress-induced effects. Our research findings demonstrated significant suppression of the expression of the *CWINV1* gene in salinized bean plants. However, the foliar application of GSE, particularly at concentrations around 6 g/L, substantially induced the expression of the *CWINV1* gene. The reduced expression of the *CWINV1* gene is linked to salinity-induced leaf senescence [81]. Conversely, the upregulation of *CWINV1* gene expression is crucial for maintaining sugar balance and vascular anatomy and subsequently influencing plant growth, development, and stress tolerance [81]. This upregulation is particularly vital for the cytokinin-mediated delay of leaf senescence, potentially enhancing plant tolerance to various stress conditions [82]. Understanding these mechanisms provides valuable insights into enhancing plant resilience and stress tolerance in agriculture.

Plants employ various metabolites to combat environmental stresses, with proline being a well-documented compound known to accumulate in cells, enhancing resistance against diverse stressors [83]. Proline synthesis can occur via the glutamate or ornithine pathways, with ornithine aminotransferase (*OAT*) playing a pivotal role in catalyzing proline biosynthesis from ornithine through the ornithine pathway. *OAT* is widely distributed in organisms and contributes significantly to stress-induced proline accumulation [84]. Studies have demonstrated that the overexpression of *OAT* enhances proline accumulation under salinity stress conditions [85]. The findings of our research indicated a significant increase in the expression of the *OAT* gene in 60-day-old salt-stressed bean plants compared to controls, directly correlating with proline accumulation. Conversely, the foliar application of GSE, particularly at a concentration of 6 g/L, markedly suppresses *OAT* gene expression in salt-stressed bean plants, leading to reduced proline accumulation. Ethylene-responsive transcription factor 1 (ERF1) belongs to the AP2/ERF transcription factor family found in plants [86]. ERF TFs play a significant role in the response to abiotic stress [87], the transduction of hormonal signals [88], and in regulating the development of the abscission zone [87]. Similarly, ethylene-responsive transcription factors (ERFs) are pivotal in conferring tolerance to various environmental stresses by modulating the expression of stress-responsive genes [88]. The overexpression of *ERF1* has been observed to result in the accumulation of proline [88]. This decrease in *OAT* and *ERF1* expressions, attributed to GSE, corresponded with diminished proline levels, likely due to the antioxidant properties and scavenging of reactive oxygen species (ROS) offered by natural plant extracts [48].

## 4. Materials and Methods

### 4.1. Preparation of Grape Seed Extract

The grape (*Vitis vinifera* L.) seeds were procured from local markets, thoroughly washed, and subsequently dried in an oven at 60 °C. The dried seeds were finely ground and the powder was mixed with distilled water and left overnight. Four different concentrations of aqueous grape seed extract were prepared: 2 g/L, 4 g/L, 6 g/L, and 8 g/L.

### 4.2. Characterizations of Grape Seed Extract Using GC–MS Spectrometry

To identify the bioactive components of *Vitis vinifera*, an ethanolic seed extract was utilized. An amount of 1 g of dried seeds was ground into a fine powder, mixed with distilled water, and left overnight. It was then put in a rotary evaporator at 45 °C to remove water from the extract. After that, it was dissolved in ethanol to estimate the bioactive compounds in the grape water extract and using GC–MS.

The 1 µL of ethanolic extract was introduced into the GC mass spectrometer (Perkin Elmer model: Clarus 580/560S, Waltham, MA, USA). The chemical composition was determined based on GC retention time (RT) on a capillary column. The resulting mass spectra were compared with standard compounds (50 to 620 Da).

For phytochemical analysis, grape seeds were digested using an HNO_3_:H_2_O_2_ mixture (5:3, *v*/*v*) to detect nitrogen (N), phosphorous (P), calcium (Ca), and magnesium (Mg) levels. The N content was estimated using the Rochelle reagent colorimetric assay, while the P content was determined using the molybdenum blue technique against standard calibration curves following the protocol of Allen et al. [89]. Ca and Mg levels were measured using inductively coupled plasma-optical spectroscopy (Polyscan 61E, Thermo Jarrell-Ash Corp., Franklin, MA, USA). Additionally, the total antioxidant capacity was assessed in the grape ethanolic extract (How you did? Reference needed or include the procedure).

### 4.3. Experimental Design

Seeds of faba beans (*Vicia faba* L. cv. Giza-716) were acquired from Gemmeiza Agriculture Research Station in El-Garbia, Egypt. The seeds were sterilized with 1% NaClO for 5 min, followed by thorough washing with distilled water. Five sterilized seeds were planted in each pot filled with 5 kg of air-dried soil (sand:clay, 3:1 *v*/*v*). The pots were irrigated daily with tap water until full germination. On the 14th day after sowing, germinated faba bean seedlings were exposed to two conditions: distilled water (+ve control) and 150 mM NaCl (−ve control). Subsequently, all salinized pots were subjected to five treatments: (1) irrigation solely with 150 mM NaCl (salinity), (2) foliar spray with 2 g/L of aqueous GSE, (3) foliar spray with 4 g/L of aqueous GSE, (4) foliar spray with 6 g/L of aqueous GSE, and (5) foliar spray with 8 g/L of aqueous GSE. The spray volume used was 30 mL per pot for each foliar application of the respective GSE concentrations. The foliar sprays were administered twice per week after the plants were exposed to 150 mM NaCl salinity stress at 14 days after sowing. The study was conducted in a completely randomized design with three replications per treatment group, each pot containing five seeds. After a period of 60 days, the plants were harvested to assess various growth parameters, such as the shoot length (cm) and root length (cm). Additionally, the fresh weight (g) and dry weight (g) of the entire plant were measured to assess its growth performance. In addition, some biochemical parameters, such as photosynthetic activity, osmo-protectant levels, and antioxidants, were estimated.

### 4.4. Measurement of Photosynthetic Activity (Fv/Fm)

Mature morphologically similar leaves were dark-adapted for 30 min before measurements. Fluorescence of dark-adapted leaves was recorded with a portable pulse amplitude modulation (PAM) fluorometer (PerkinElmer, Waltham, MA, USA). Three replicated samples of each treatment were recorded. The dark-adapted leaves were secured in a leaf clip to ensure consistent angles of incidence between the fiber-optic arm of the fluorometer and the leaf surface. The measurement of fluorescence on the adaxial leaf surface was essentially recommended, as described by [90].

### 4.5. Measurement of Oxidative Stress Markers

The malondialdehyde (MDA) content was determined using the thiobarbituric acid (TBA) reaction method, as per Heath and Packer [91]. Absorbance readings were taken at 532 nm and 600 nm. MDA content was calculated using an extinction coefficient of 1.55 µM^−1^ cm^−1^ and was expressed as nmol/g fresh weight. Hydrogen peroxide content was estimated following the protocol of Velikova et al. [92]. Absorbance measurements were taken at 390 nm, and H_2_O_2_ levels were calculated using an extinction coefficient of 0.28 µM^–1^ cm^–1^ and expressed as µmol/g fresh weight.

### 4.6. Measurement of Osmo-Protectants

Proline content was determined according to [93]. Absorbance readings were taken at 520 nm, and proline content was expressed as mg/g dry weight based on a standard graph prepared using proline. Glycine betaine content was estimated in watery extracts of dry plant powders following the method outlined by Grieve et al. [94]. Absorbance was measured at 365 nm, and glycine betaine content was calculated as mg/g dry weight.

Total amino acids were quantified in prepared ethanolic extracts with a spectrophotometer, using the method described by Lee and Takahashi [95]. The concentration of total amino acids was expressed as mg/g dry wt. Total soluble proteins were extracted from dried plant tissues according to Naguib et al. [96] and quantitatively measured following Bradford [97]. Absorbance readings were taken at 595 nm, and protein content was calculated as mg/g dry weight using a calibration curve prepared with bovine serum albumin (BSA) protein.

Total soluble sugar content was determined in borate buffer extracts, following the method by Dubois et al. [98]. Absorbance readings were taken at 490 nm, and total soluble sugar content was calculated as mg/g dry weight using a prepared calibration curve.

### 4.7. Measurement of Antioxidants

Total antioxidant capacity (TAC) was assessed using the phosphomolybdenum method [99]. TAC reagent preparation involved sulfuric acid (0.6 M), ammonium molybdate tetrahydrate (4 mM), and sodium phosphate dibasic solution (28 mM). Methanolic extract mixed with TAC reagent was boiled for 90 min, followed by measurement of absorbance at 765 nm post-cooling.

### 4.8. Determination of Peroxidase, Ascorbate Peroxidase and Polyphenol Oxidase

Antioxidant enzymes (peroxidase, ascorbate peroxidase, and polyphenol oxidase) were extracted according to Grace and Logan [100], in which fresh leaves were ground and homogenized in potassium phosphate buffer (50 mM, pH 7.0) containing 4% PVP, 2% glycerol, and 0.1 mM EDTA. Homogenates were centrifuged at 15,000 rpm for 30 min at 4 °C, and the supernatants were used to analyze enzyme activities. Peroxidase (POD) activity was assessed according to Kato et al. [101]. The activity of POD was calculated by using the extinction coefficient of 26.6 mM^−1^ cm^−1^ at 470 nm. Ascorbate peroxidase (APX) activity was assayed according to the method recommended by Nakano et al. [102]. Polyphenol oxidase activity (PPO) was assayed according to the method described by Zhou et al. [103], the absorbance was measured at 420 nm using the extinction coefficient of 26.40 M^−1^ cm^−1^, and the enzyme activity was expressed as µM/g fresh weight min^−1^.

### 4.9. Expression Profiling of Selected Genes

Total RNA was extracted from faba bean leaves using the RNeasy Plant Mini Kit (Qiagen), following the manufacturer’s instructions. The RNeasy Plant Mini Kit includes QIAshredder spin columns for homogenizing and filtering viscous plant lysates and RNeasy spin columns for purifying up to 100 μg of high-quality RNA using silica-membrane technology. Complementary DNA (cDNA) was synthesized via reverse transcription of RNA in a 20 μL reaction volume using a PTC-100™ thermocycler (MJ Research, Waltham, MA, USA). The reaction conditions involved an initial enzyme activation cycle at 42 °C for 1 h, followed by an enzyme inactivation cycle at 95 °C for 5 min. The qRT-PCR was performed in triplicate using the SYBR Green PCR Master Mix (Fermentas, Waltham, MA, USA). Each reaction contained a 25 μL mixture with specific primer pairs (Table 3), and data were collected during the extension step. The reactions were carried out on a Rotor-Gene 6000 system (QIAGEN, Germantown, MD, USA). The GAPDH (glyceraldehyde 3-phosphate dehydrogenase) gene served as the reference gene, as described by [104]. Quantification and calculation of the relative expression of the studied genes were performed following the method outlined by Livak and Schmittgen [105].

### 4.10. Statistical Analysis

The results are presented as the mean of three replicates ± standard error (SE). Differences between treatments for the different measured variables were tested by one-way analysis of variance (ANOVA), followed by Duncan’s test, with significant differences found (*p* < 0.05) using XLSTAT software (version 2014.5.03).

## 5. Conclusions

Overall, the findings of this study highlighted the potential of GSE as a natural, sustainable, and effective mitigation strategy for soil salinization. The comprehensive analysis of growth parameters, phytochemical composition, and molecular responses elucidated the multifaceted benefits of GSE in promoting plant resilience by modulating osmolyte levels, enhancing antioxidant capacity, and regulating gene expression associated with photosynthesis and stress responses of bean plants under salinity stress conditions. Furthermore, our findings provide valuable insights into the mechanisms underlying the protective effects of grape seed extract, paving the way for further research and the optimization of application methods in sustainable agriculture practices. Future recommendations emphasize the need for the comprehensive evaluation and optimization of grape seed extract (GSE) application to maximize its potential as an effective salinity mitigation strategy. Assessing the long-term efficacy and scalability of GSE across diverse crop species, agroecosystems, and stress conditions is crucial to determine its broad applicability in sustainable agriculture practices. Furthermore, investigating synergistic effects by combining GSE with other natural or synthetic compounds could lead to the development of integrated strategies for enhanced stress tolerance. Conducting dose-response studies and optimizing application methods, such as foliar sprays or soil amendments, will be essential to maximize the efficacy and cost-effectiveness of GSE application. Additionally, employing advanced omics techniques, including transcriptomics and metabolomics, can elucidate the molecular mechanisms underlying GSE’s protective effects, enabling the development of more targeted and efficient mitigation strategies tailored to specific crop species and environmental conditions.

## Figures and Tables

**Figure 1 plants-13-01596-f001:**
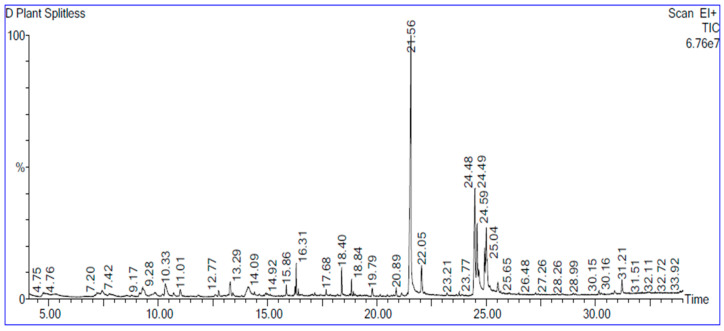
GC-MS spectral chromatogram of grape ethanolic seed extract.

**Figure 2 plants-13-01596-f002:**
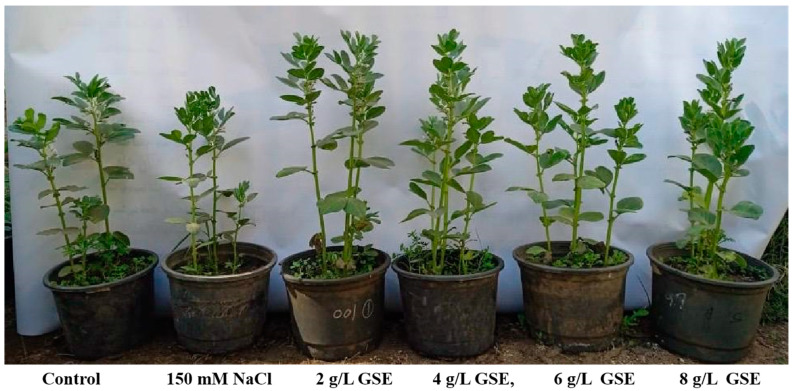
Faba bean plants under different treatments: control; 150 mM NaCl (salinity); foliar spray with 2 g/L of aqueous GSE; foliar spray with 4 g/L of aqueous GSE; foliar spray with 6 g/L of aqueous GSE; and foliar spray with 8 g/L of aqueous GSE.

**Figure 3 plants-13-01596-f003:**
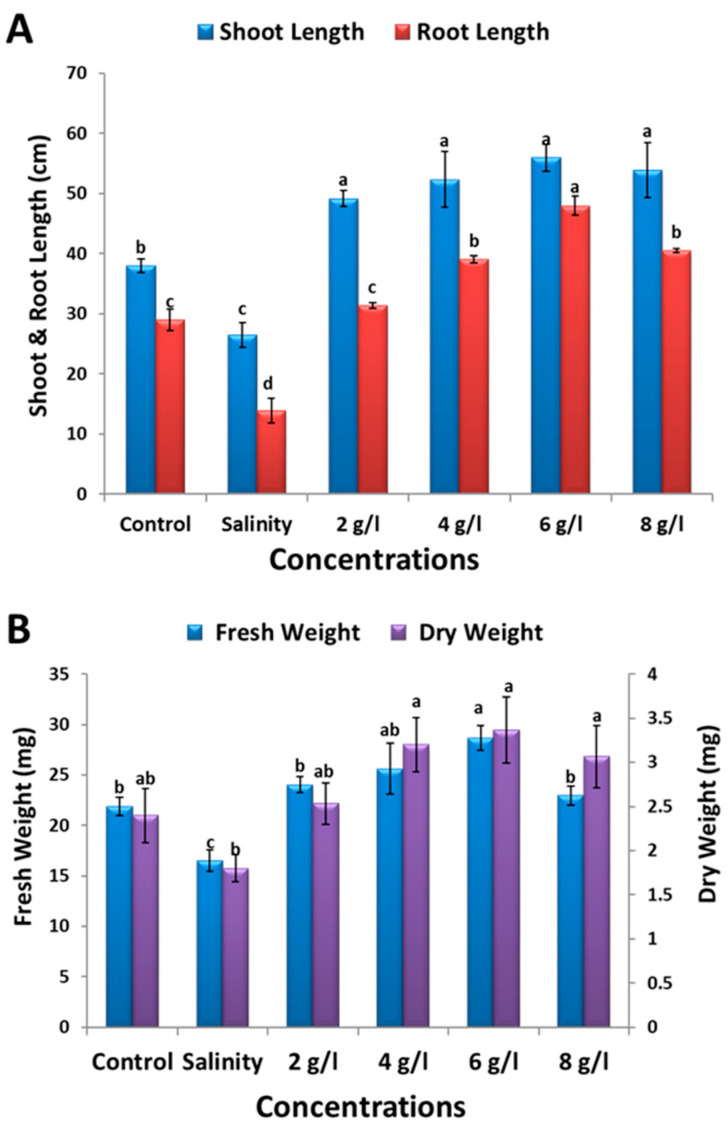
Effect of foliar spray with grape seed extract (2, 4, 6, and 8 g/L) on (**A**); shoot length, root length, (**B**); fresh weight, and dry weight of salt-stressed bean plants. Different letters denote the significant variations measured by Duncan’s multiple range test at *p* < 0.05.

**Figure 4 plants-13-01596-f004:**
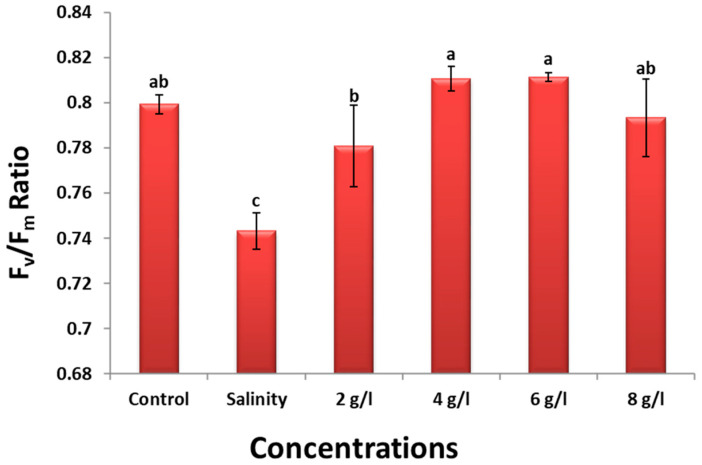
Effect of foliar spray with grape seed extract (2, 4, 6, and 8 g/L) on Fv/Fm ratio of salinized bean plant. Different letters denote the significant variations measured by Duncan’s multiple range test at *p* < 0.05.

**Figure 5 plants-13-01596-f005:**
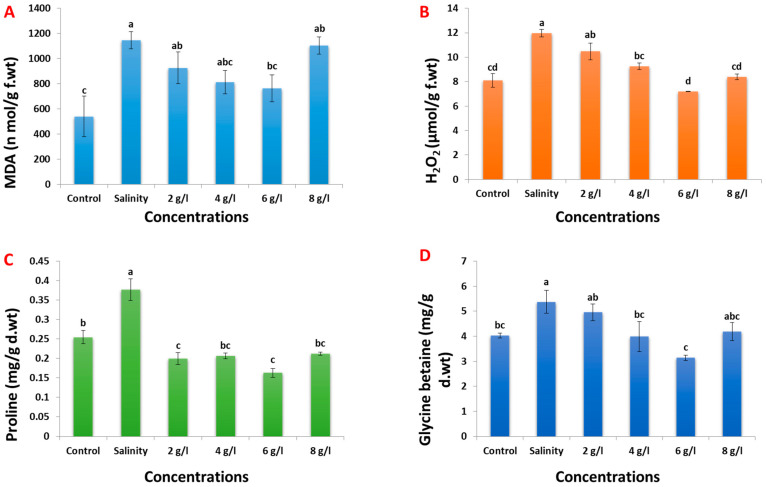
Effect of foliar spray with grape seed extract (2, 4, 6, and 8 g/L) on MDA (**A**), H_2_O_2_ (**B**), proline (**C**), and glycine betaine contents (**D**) of salinized bean plant. Different letters denote the significant variations measured by Duncan’s multiple range test at *p* < 0.05.

**Figure 6 plants-13-01596-f006:**
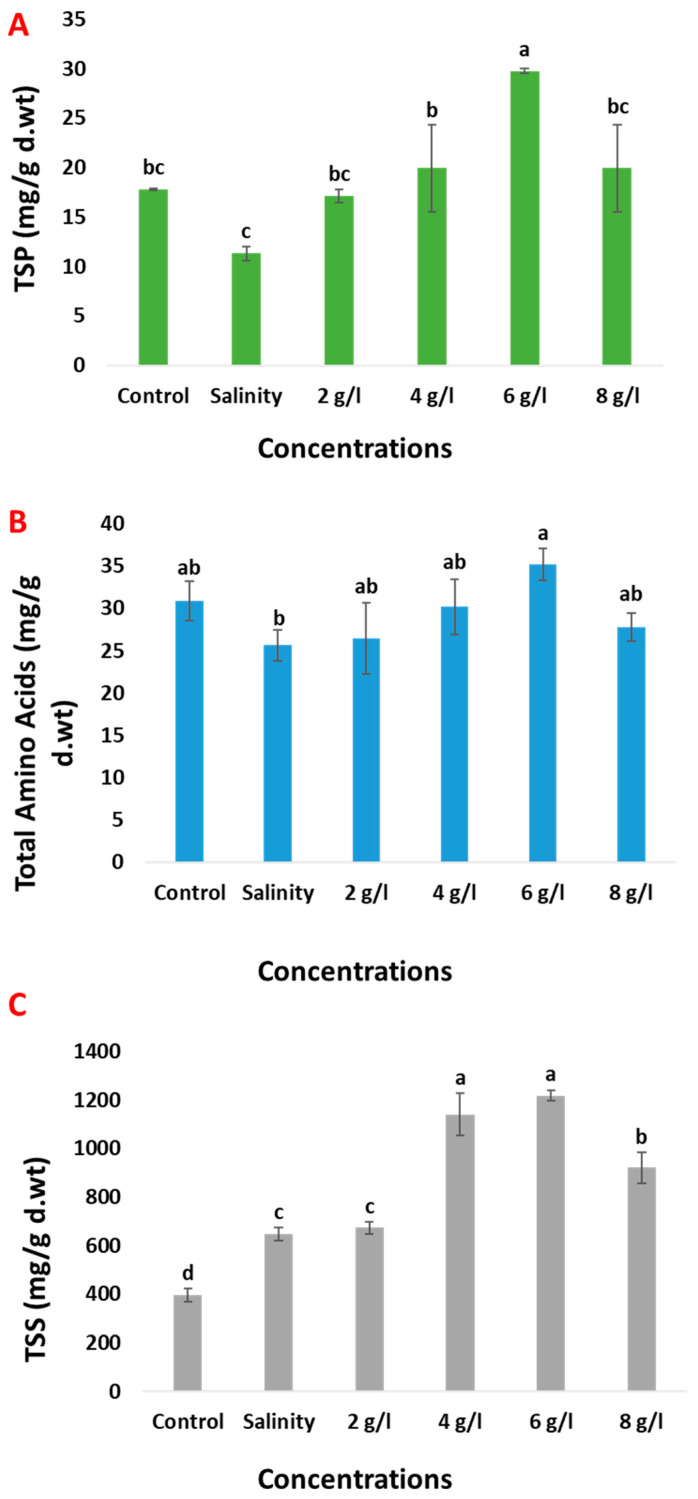
Effect of foliar spray with grape seed extract (2, 4, 6 and 8 g/L) on (**A**); TSP, (**B**); TAA, and (**C**); TSS content (mg/g d.wt) of salinized bean plant. Different letters denote the significant variations measured by Duncan’s multiple range test at *p* < 0.05.

**Figure 7 plants-13-01596-f007:**
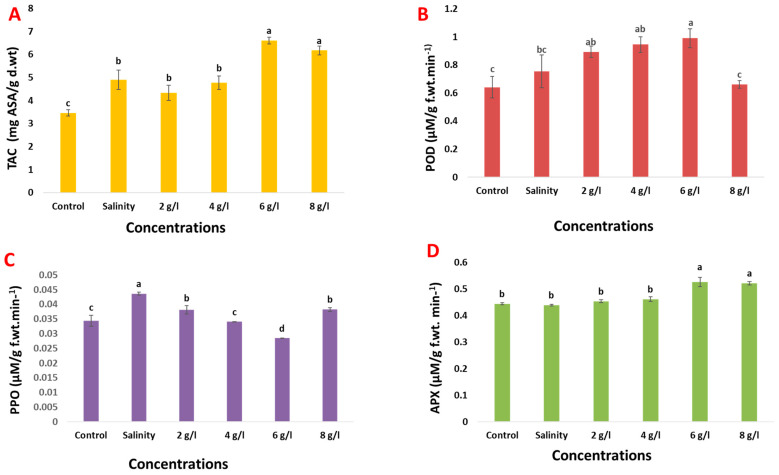
Effect of foliar spray with grape seed extract (2, 4, 6 and 8 g/L) on (**A**) TAC, (**B**) POD, (**C**) PPO, and (**D**) APX (µM/g f.wt.min^−1^) of salinized bean plant. Different letters denote the significant variations measured by Duncan’s multiple range test at *p* < 0.05.

**Figure 8 plants-13-01596-f008:**
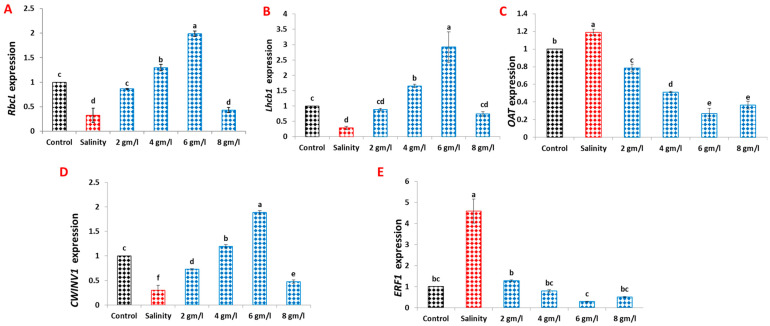
Effect of foliar spray with grape seed extract (2, 4, 6, and 8 g/L) on relative gene expression of RbcL (**A**), Lhcb1 (**B**), OAT (**C**), CWINV1 (**D**), and ERF1 (**E**) of salinized bean plants. Different letters denote the significant variations measured by Duncan’s multiple range test at *p* < 0.05.

**Figure 9 plants-13-01596-f009:**
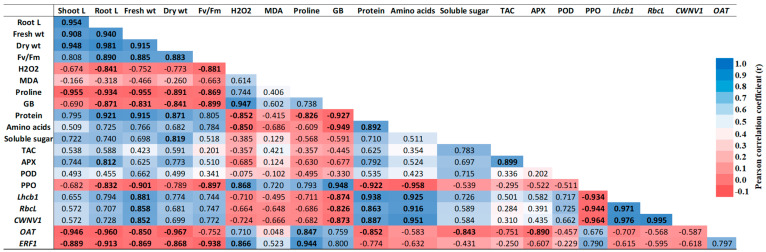
Pearson correlation matrix of different morpho-physiological and biochemical traits under salinity stress.

**Table 1 plants-13-01596-t001:** Phytochemical components of grape ethanolic seed extract identified by GC–MS.

NO	Compound Name	Molecular Formula	RT	Area %
1.	Hexanoic acid	C_6_H_12_O_2_	4.799	1.690
2.	Octanoic Acid	C_8_H_16_O_2_	9.280	2.065
3.	2-Furancarboxaldehyde, 5-(hydroxymethyl)-	C₆H₆O₃	10.331	2.203
4.	à-D-Glucopyranoside, O-à-D-glucopyranosyl-(1.fwdarw.3)-á-D-fructofuranosyl	C_18_H_32_O_16_	14.122	3.056
5.	n-Hexadecanoic acid	C₁₆H₃₂O₂	21.560	26.350
6.	Hexadecanoic acid, ethyl ester	C_18_H_36_O_2_	22.051	2.263
7.	9,12-Octadecadienoic acid (Z,Z)-	C_18_H_32_O_2_	24.492	7.442
8.	Hexadecenoic acid, Z-11-	C_16_H_30_O_2_	24.587	3.754
9.	Linoleic acid ethyl ester	C_20_H_36_O_2_	24.932	2.586
10.	Octadecanoic acid	C_18_H_36_O_2_	25.007	6.560

**Table 2 plants-13-01596-t002:** Quantitative nutritional contents of Vitis vinifera seed extract.

Parameters	Results	Unit
N ions	15.13 ± 0.4	mg/g DM
P ions	1.4 ± 0.4	mg/g DM
Ca ions	18.29 ± 0.1	mg/g DM
Mg ions	5.34 ± 0.1	mg/g DM
Total antioxidant capacity	9.85 ± 0.6	mg ASA/g DM

**Table 3 plants-13-01596-t003:** Primers for real time q-PCR study for salt stress.

Gene Name	Abbreviation	Forward (F) and Reverse (R) Primer 5′–3′
Reference gene	*GAPDH*	F: 5′-AAGGTTATCAACGACAGGTTTG-3′ R: 5′-ATACCCTTAAGCTTGCCTTCTG-3′
Chlorophyll a/b-binding protein of LHCII type 1-like (LOC114182484)	*Lhcb1*	F: 5′-GGCTTTTGCTGAGTTGAAGG-3′ R: 5′-GTAAGCCCAGGCATTGTTGT-3′
Ribulose bisphosphate carboxylase large chain-like. Number in gene bank (X01167)	*RbcL*	F: 5′-CTTGGTACCATCCAACCAATTCA-3′ R: 5′-GCTTGGAACCCAACCTTTGC-3′
Cell wall Invertase I Number in gene bank (Z35162)	*CWINV1*	F: 5′-GGGTTGGACCGTTTGGACTT-3′ R: 5′-CACGCCCGATTAAAACCATACT-3′
Ornithine aminotransferase (loc107482012)	*OAT*	F: 5′-GAATACTGGCGCTGAAGGTGTG-3′ R: 5′-AGATGGCCAGGCAATAAAGGAC-3′
Ethylene-responsive transcription Factor 1 Number in gene bank (EU543659)	*ERF1*	F: 5′-TGCTGCTTTTCATTTTCGTG-3′ R: 5′-AGGCGCTGTAAGAGGCATAG-3′

## Data Availability

The original contributions presented in the study are included in the article, further inquiries can be directed to the corresponding author.

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
