# Peer review of "Phytochemical Profiling and Bioactive Potential of Grape Seed Extract in Enhancing Salinity Tolerance of Vicia faba"

_plants, 2024, doi:10.3390/plants13121596_

Round 1

Reviewer 1 Report

Comments and Suggestions for Authors

Mitigating the negative effects of salt stress or enhancing salt tolerance in plants is a continuing research interest. This manuscript explores for the first time the possibility of grape seed extract (GSE) to alleviate salt stress effects in crops. This manuscript systematically investigates the effects of GSE on the physiology and biochemistry of fava bean, and the results confirm the effectiveness of GSE in agricultural applications. Here are some suggestions/questions:

1. The introduction needs to add other extracts in alleviating plant salt stress as well as relevant studies on the composition of grape seed extract and its application.

2. The authors need to elaborate on the method of preparation of grape seed extract. Why was the active composition of the grape ethanolic extract determined when the study used the grape water extract? Similarly, why grape water extract and grape ethanol extract were used for determination of betaine content and total amino acid content respectively.

3. Line 361, Methods of determination of antioxidant capacity of grape ethanol extract need to be clarified.

4. Section 4,3, What are the spray volumes of different concentrations of GSE? Spraying once a day? The methods for each treatment likewise need to be described in detail.

5. Line 393, 133, 135, subscripts for hydrogen peroxide.

6. Line 431, RNase Mini Kit details need to be specified.

7. Line 95, GC-MS.

8. Line 95, Pictures of fava beans under each treatment are recommended.

9. It is recommended that pictures of fava beans under each treatment be added.

10. Line 120, 6% may be wrong.

11. Line 122, the data increase here is relative to salt treatment or control treatment.

12. Line 141, it should be 30% increase instead of 1.3-fold, similarly, line 143, it should be 10%, 40%, 70% and 30% decrease.

12. Line 150, there was no significant difference in TSP and TAA in salt and control treated beans. Same, no significant change in POD activity.

13. Significant results need to be added to Figure 8, currently only correlation coefficients are shown.

14. Why is MDA content positively correlated with TAC and APX?

15. It is recommended that the authors analyse the content of actives in different concentrations of grape seed extracts.

Reviewer 2 Report

Comments and Suggestions for Authors

The study provides “phytochemical profiling and bioactive potential of grape seed extract in enhancing salinity tolerance of Vicia faba L”. The study is well designed. However, there are some limitations which must be addressed.

 The abstract provides a good overview of the study, but more details on the experimental design, treatments, and statistical analyses would be beneficial in the main text.

Compare the results with other well-known antioxidants or standards to provide context for the observed values.

Line 27-30 abstract provides specific information.

Paragraph 2 of the introduction, provides more information specifically regarding Vicia faba. Mechanism, tolerance level, and significance.

The growth parameter measurements (shoot length, root length, fresh weight, dry weight) are relevant, but it would be useful to include information on the experimental design, such as the number of replicates, randomization, and statistical analyses used.

Line 54-55 must be cited with recent studies https://doi.org/10.1007/s10725-024-01128-y, https://doi.org/10.3390/genes13101699

Provide more information on the specific methods used and any quality control measures taken.

The discussion section could be expanded to better contextualize the findings within the existing literature on salinity stress mitigation and the potential mechanisms by which GSE exerts its effects.

Line 84 Write “Figure 1” with small f.

It would be beneficial to include a discussion on the potential practical applications of using GSE as a natural approach to enhance salt tolerance in agricultural settings, considering factors such as cost, availability, and scalability.

While the gene names (RbcL, Lhcb1, CWINV1, OAT, and ERF1) are provided, there is no information about the specific functions or roles of these genes in the context of salinity stress and the potential mechanisms by which GSE modulates their expression.

The text does not mention whether the gene expression analysis accounted for potential confounding factors, such as developmental stage, tissue type, or other environmental conditions.

Provide future recommendations in conclusion

Round 2

Reviewer 1 Report

Comments and Suggestions for Authors

The author has responded to all my comments and made careful revisions to the paper.